# Subthreshold Autism and ADHD: A Brief Narrative Review for Frontline Clinicians

**DOI:** 10.3390/pediatric17020042

**Published:** 2025-04-03

**Authors:** Michael O. Ogundele, Michael J. S. Morton

**Affiliations:** 1Department of Community Paediatrics, King’s Mill Hospital, Sutton in Ashfield, Nottingham NG17 4JL, UK; 2School of Health & Wellbeing, University of Glasgow, Clarice Pears Building (Level), 90 Byres Road, Glasgow G12 8TB, UK; michael.morton@glasgow.ac.uk

**Keywords:** subthreshold, neurobehavioral disorders, childhood, traits, ADHD, ASD

## Abstract

**Background:** Epidemiological studies have shown that neurodevelopmental disorders (NDDs), such as autism spectrum disorder (ASD) and attention deficit/hyperactivity disorder (ADHD) are more prevalent in the general childhood population, compared to cases that are formally diagnosed in clinical cohorts. This suggests that many children and youths have NDD which are never diagnosed clinically, causing impairments in some domains of their daily life. There is increasing recognition of the concept of a “subthreshold” condition, sometimes used to describe the presence of potentially impairing variations in the neurodevelopmental profile that do not meet criteria for a diagnosis. The aim of this narrative review is to appraise the published literature about common themes regarding subthreshold conditions in relation to autism and ADHD, identifying any practical lessons that may be applicable to frontline neurodevelopmental clinicians. **Methods:** We searched electronic databases including PMC and PubMed using various combinations of keywords, including “Subthreshold”, “subclinical”, “neurodevelopmental”, “childhood”, “ADHD” and “ASD”. **Results:** The identified themes include definitions, prevalence, assessment tools, lifetime impairments, NDD classification models, management, raising public awareness, and future research directions. **Conclusions:** The authors propose that a “subthreshold condition” should be recorded when NDDs do not meet current diagnostic criteria if there is evidence of significant, persisting impairment in at least one setting.

## 1. Introduction

The category of “neurodevelopmental disorders” (NDDs) includes a number of long-term conditions caused by dysfunctions of the brain and or neuromuscular system leading to functional limitations in multiple domains. According to the *Diagnostic and Statistical Manual of Mental Disorders (5th edition)* (DSM-5), NDD conditions include common neurobehavioral disorders with onset in early childhood, such as attention deficit/hyperactivity disorder (ADHD), autism spectrum disorder (ASD), other neurodevelopmental conditions (such as tic disorders, developmental language, and coordination disorders), and intellectual disability. Their co-occurrence with a wider range of emotional, behavioral, and intellectual disorders is the norm. This has led us to propose a wider umbrella terminology of NDEBIDs—neurodevelopmental, emotional, behavioral, and intellectual disorders [1]. The term “intellectual disorders” includes mild to severe disorders of intellectual development (the *International Classification of Diseases* (11th edition-ICD-11) and intellectual disability (DSM-5). Gillberg et al. [2] have emphasized the importance of a comprehensive assessment of CYP referred for a range of NDDs and general mental health concerns, due to the high degree of comorbidity and co-existence of these related conditions.

Children and young people (CYP) with NDD may require assessment and treatment from a wide of range of multi-agency and multidisciplinary professionals including psychiatrists, psychologists, pediatricians, speech and language therapists, and occupational therapists, together with social care, voluntary sector, and educational agencies [3]. Fragmentation of treatment and support services is unfortunately a common experience of patients and their carers [4]. There are strong arguments in favor of greater integration between mental health, physical and neurodevelopmental pediatric services. For example, CYP with neurodevelopmental and intellectual disorders have a three- to four-fold increase in prevalence of co-occurring mental disorders into adulthood [5]. The current picture of integration between community child health (CCH) and child and adolescent mental health (CAMH) is complex and mixed [6].

Epidemiological studies have shown that these conditions are more prevalent in the general childhood population, compared to the cases that are formally diagnosed in clinical cohorts [1]. This suggests that many more CYP have NDDs who are never diagnosed clinically, but they may still experience impairments in different domains of their daily life. Such conditions might be described as “subclinical”, as they have not come to the point of receiving a diagnosis. Associated with this observation and sometimes overlapping with it is the concept of “subthreshold” diagnoses, which include mild/atypical symptoms, gender-specific features, particular behavioral manifestations, and personality traits associated with various typical NDDs [7]. These additional components of the neurodevelopmental profile can be of great importance in a clinical setting, meaning that the needs of children with the same NDD diagnosis may differ considerably. One problem with this approach is the necessity to consider how best to define “subthreshold conditions” [3]. To establish a definition of subthreshold conditions most reliably the term can be restricted to the identification of distinctive developmental traits or patterns of behaviour that do not meet the full criteria for a formal clinical diagnosis. It may seem counterintuitive to create a second, lower threshold that must be met to identify a subthreshold disorder. An alternative approach in considering the individual case is to explore the relation between an individual’s neurodevelopmental profile and any presenting impairment, considering the psychopathological context as a whole [8].

Apart from the identification of traits in individuals presenting with impairments, there is also a need to address the possible impact of subthreshold conditions as risk factors for future disorders in individuals who are not currently impaired. Identification of subthreshold conditions in children with no existing diagnosis raises concerns about the extension of labelling of difference to a wider group with the ensuing risks of medicalization of conditions that lie within the normal spectrum [9]. This caution carries weight when it is acknowledged that in the absence of a diagnosis there is a dearth of evidence to validate interventions, which limits the options available to clinicians and educators seeking to alleviate the impact of impairing conditions. This is particularly pertinent when many systems currently lack resources to provide evidence-based care for children with established diagnoses. Recent research nevertheless suggests that the importance of identifying subthreshold conditions may outweigh any disadvantage. CYP as well as adults may miss out on support strategies and provisions, despite severe functional impairment from their behavioral traits, simply because they have not met the criteria for a diagnosis. Furthermore, many people suffer because of multiple significant NDD traits, each of which do not individually meet the threshold for a specific diagnosis. In addition, subthreshold traits associated with NDDs are now recognized to be risk factors for significant negative mental health consequences due to lack of appropriate support and necessary interventions.

Many NDD conditions are increasingly being identified as part of a more complex multi-dimensional heterogeneous spectrum, for which the classical single diagnosis paradigm is unsuitable to identify the degree of their impairment and clinical presentation. For example, there is a continuum of symptomatology between manifestations of eating disorders and autism spectrum disorders [10], as well as an overlap between ASD and ADHD symptoms [11]. Some authors have also advocated that mental disorders be recognized as a system rather than as a category. This model proposes that future research might explore the dynamics of system change (e.g., abrupt vs. gradual psychosis onset), the factors to which these systems are most sensitive (e.g., interpersonal dynamics and neurochemical change), and the individual variability in system architecture and change [12].

Taxometric research using systematized approaches and mathematical procedures has tried to identify how best to classify and interpret differences in human characteristics and psychopathology. A recent meta-analysis of 317 findings from 183 taxometric published articles has shown that the majority of psychological differences and psychopathology symptoms have a continuous distribution (dimensional) rather than strictly categorical classifications [13]. This supports the notion that variations in symptoms of NDEBIDs are best understood as differences in degree (dimensional) with quantitative variations along a continuum, rather than discrete kinds (categorical or typological). This has implications for several areas of clinical practice including research, assessment (dichotomous diagnosis vs. quantitative measurement) and classification (categorical vs. dimensional taxonomy). The very few exceptions that may be considered to be more suitable for categorical classification in childhood include autism and suicide risk [13]. Thapar et al. [3] argue for a pragmatic approach, recognizing that neurodevelopmental disorders are more than a set of diagnostic criteria, so that the complex interaction of subthreshold conditions with NDDs is acknowledged in both research and clinical practice.

The aim of this narrative review is to appraise the general published literature about common themes regarding subthreshold NDD, focusing mainly on ADHD and ASD. We aim to identify practical lessons that may be applicable as effective public health measures and sometimes within a clinical setting.

## 2. Materials and Methods

A search of the published literature from electronic databases including PMC and PubMed using various combinations of keywords, including “Subthreshold”, “subclinical”, “neurobehavioral disorders”, “neurodevelopmental”, “childhood”, “ADHD”, “ASD”, “ODD”, and “CD”. We identified relevant articles published between 2000 and June 2024, and any other earlier articles identified in the references were also reviewed. This is a narrative review of the extant literature and was not designed to be a systematic review. This topic lends itself to a narrative rather than systematic review because of the broad and interdisciplinary nature of the topic. There are still no standard agreed definitions or gold-standard diagnostic criteria for identifying “subthreshold conditions”. The methodology of systematic reviews could not synthesize the very heterogynous extant literature. A narrative review enables us to summarize existing knowledge and provide a comprehensive background on the topic.

The following themes were identified: definitions, assessment tools, prevalence, risk factors and neurobiology, lifetime functional impairments, review of the NDD classification conceptual models, management, raising public awareness, and future research directions. The chosen search strategy failed to identify the range of research in the rapidly moving field of the genetics of neurodiversity, and a decision was taken to accept this. This area is well reviewed elsewhere, such as by Leblond et al. [14].

## 3. Results

### 3.1. Definitions

“Neurodevelopmental disorders” comprise a group of congenital or acquired long-term conditions, generally apparent before puberty, which are attributed to disturbances of the brain and or neuromuscular system and which create functional limitations, including ADHD, ASD, tic disorder/Tourette’s syndrome, developmental language disorders, and intellectual disabilities. These NDDs often cluster together with each other and commonly co-occur with several other emotional, sleep, and behavioral problems [2,15,16].

Subthreshold traits refer to a pattern of symptoms which do not reach the criteria for an individual NDD diagnosis and which are commonly found in association with NDDs [3]. Other authors also refer to subthreshold disorders as conditions with relevant psychiatric symptoms which do not meet the full criteria for a clinical diagnosis [17]. Features that might not meet full diagnostic criteria may include the number and frequency of symptoms, the presence of symptoms/impairment in one but not in other settings, or discrepancies between multi-source observations [18]. There is a particular problem with regard to the situation-specific expression of traits of a disorder as these may be wrongly interpreted as neurodevelopmental in origin while an environmental cause may be overlooked. The literature is not clear on the validity of the subthreshold concept where the criterion of age of onset in childhood is not met. Subthreshold conditions are also considered as particular personality traits which may be responsible for the “uniqueness” of every human being, in varying degrees of intensity [19]. The use of the term “subthreshold disorder” can imply that specific diagnostic criteria have been considered, and the term is sometimes used after a clinical assessment as if it were a recognized diagnosis. However, the language of “traits” may be more appropriately used when referring to specific features, such as “autistic traits”. The term “subthreshold condition” acknowledges a degree of wider variation and complexity of symptoms and may be more relevant in a public health context. Research definitions of subthreshold conditions are often derived from scores in the assessment tools used and there is no single consensus approach. Subthreshold conditions can be identified in CYP referred to clinical settings who have been assessed with standard classification manuals, such as the DSM-5 or ICD-11, but their symptoms and degree of impairments do not fulfil all the criteria for a diagnosis (Figure 1).

### 3.2. Assessment Tools

There are no universally accepted gold-standard assessment tools for all subthreshold conditions. Some validated tools have been developed and used for the assessment of subthreshold autism traits. These include the Structured Clinical Interview for DSM-5 (SCID-5), the Adult Autism Subthreshold Spectrum (AdAS Spectrum) for autistic traits [20,21,22], the Camouflaging AT Questionnaire [23], and the Subthreshold Autism Trait Questionnaire (SATQ) [24]. Subthreshold ADHD and other psychiatric disorders are often diagnosed when the full criteria of either the DSM-5 or ICD-11 classifications are not met or based on subthreshold scores on the standard assessment tools, such as the Montgomery–Åsberg Depression Rating Scale (MADRS), the State–Trait Anxiety Inventory (STAI) [25], and the Kiddie-SADS Present and Lifetime Version (K-SADS-PL) [26].

Some psychological assessment tools, such as the Social Responsiveness Scale (SRS), a quantitative behavioral measure of autistic traits, show a continuous distribution when applied to normal childhood population, indicating no evidence of a natural gap that could differentiate children diagnosed with ASD from subthreshold or unaffected children [27]. In describing their Japanese study, Kamio et al. caution that where the paradigms for categorical case assignment are superimposed on a continuous distribution, this can result in substantial variation in prevalence estimation, especially when the measurements used are not standardized for a given population (i.e., by gender, informant, culture, etc.) [27]. Nevertheless, such assessment tools, when used appropriately, may lend themselves to the creation of a range of scores falling below the diagnostic threshold which, after validation of the concept, might be used to define a subthreshold condition.

### 3.3. Prevalence

Autistic or ADHD traits are commonly recognizable among the general population [28], but estimates of prevalence are limited by the lack of a universally accepted approach to their identification. They are more prevalent among CYP and adults with one or more NDEBID conditions, and among their parents and siblings [29,30]. For example, autism-specific social and communication difficulties (subthreshold autistic traits) have been reported with a prevalence of 10–17% among CYP with OCD, not related to OCD severity [31]. Comorbidity and co-occurrence in clusters is the norm for NDEBID conditions [2,16,32]. Several subclinical comorbidities are also common among CYP with ASD [33].

The exact prevalences of subthreshold ADHD is unknown due to the limitations of research studies. Balázs & Keresztény [17] reviewed 18 articles and found a wide range of prevalences of subthreshold ADHD (0.8–23.1%). They attributed this to wide variations in the definitions and assessment methods utilized by various authors. A qualitative review (*n* = 15 manuscripts) and meta-analysis (*n* = 9 manuscripts) of the published literature estimated the mean prevalence of subthreshold ADHD to be 17.7% among clinically referred and non-referred children [34]. A small study of 10–11-year-old children with ADHD and subthreshold ADHD reported population prevalence rates of 5.4% and 1.6%, respectively [35]. Twin studies among 9–12-year-old Swedish children reported a prevalence of 1.78% and 9.75% for diagnosed and subthreshold criteria of ADHD, respectively [36]. From a study of 2493 Korean elementary school children, prevalences of 5.90% (95% confidence interval = 4.74–7.06) for full syndrome and 9.00% (95% confidence interval = 7.58–10.41) for subthreshold ADHD were found [37]. A WHO study based upon retrospective adult interviews found rates of subthreshold childhood ADHD averaged at 3.7% across 18 countries, varying between 4.7% in high-income countries to 2.2% in low-/lower- and middle-income countries [38].

Table 1 presents a summary of studies showing the prevalence of subthreshold ASD and ADHD. Variation in the rates of subthreshold disorders may reflect variations in the definitions and the instruments used to conduct research. The complex co-occurrence of subthreshold disorders and diagnosed NDEBIDs is also a difficulty in assessing prevalence. The identification of individuals with subthreshold NDEBID conditions is further complicated by the absence of altered executive functions or abnormal electrophysiological activity, despite reported significant psychological impairments and comorbidity [39].

There is a difference between male: female prevalence among those with full-blown common NDEBID diagnoses compared to those with subthreshold symptoms. Subthreshold ADHD was equally prevalent in boys and girls, and more prevalent in low-income families, compared to male predominance among CYP with a full ADHD diagnosis [40].

### 3.4. Risk Factors and Neurobiology

Recent research in the genetics of neurodiversity is beyond the scope of this review. The potential importance of this topic was established in a 2015 review of twin and family genetic studies which suggested that the siblings of children with ASD and ADHD are at risk not only of clinically elevated problems in these areas, but also of subthreshold symptoms and/or subtle impairments in various neurocognitive skills and other domains of psychosocial health [41]. A UK longitudinal twin study found that the shared risk factors for full-blown de novo late-onset ADHD and late-onset ADHD with a pre-existing subthreshold condition were male sex and increased childhood conduct problems [42]. This study also found that the number of previously identified independent risk factors for ADHD positively correlated with the number of ADHD symptoms identified, so more predictors were found in individuals with a full diagnosis of ADHD in childhood than in those with a subthreshold condition. Given the similarities in etiology, it is perhaps not a surprise that in a French population sample followed up after nine years, Lecendreux et al. [43] describe the potential for subthreshold ADHD to be identified in individuals who have previously fulfilled the diagnostic criteria for ADHD in early childhood.

Limited studies with functional magnetic resonance imaging (fMRI) have suggested that the brain functional connectivity involving visual and motor networks found in young people with subclinical autistic traits is unique and different from that identified in those with full-clinical diagnosis and in non-autistics [44].

### 3.5. Lifetime Functional Impairment

Studies have confirmed that CYP assessed for NDEBIDs who did not meet all the criteria for a full diagnosis still experience significant lifetime impairment in several domains and may require ongoing additional support and monitoring for potential future psychopathology. In the prospective, longitudinal Great Smoky Mountains Study, an adult sample showed that people who experienced any childhood psychiatric disorder had a 6- to 9-times higher risk of at least one adverse outcome related to health, the legal system, personal finances, and social functioning; this compared to a 3- to 5-times higher risk of at least one adverse outcome among those with subthreshold problems [45].

A review and meta-analysis showed high risks for negative outcomes in a wide range of non-overlapping functional domains in CYP with subthreshold ADHD, compared to children with no ADHD symptoms, including significantly higher rates of family dysfunction, cognitive impairment, executive dysfunction, interpersonal and school deficits, temperament problems, psychiatric comorbidity, and juvenile delinquency [34]. This finding is supported by a multicenter cross-sectional study in Italy involving 440 adolescents that has shown that those with subclinical ADHD also manifest the same range of unhealthy lifestyle behaviour as those with ADHD. These behaviors include energy drinks/alcohol consumption, altered mindful eating, impaired quality of sleep, and problematic technology use [46]. Also, in a Chinese clinic-based study subthreshold ADHD was associated with the same level of comorbid internalizing behaviors and several aspects of executive functioning as in those with full diagnosis, although externalizing symptoms and caregiver strain were at levels between those found in controls and ADHD cases [47]. This emphasizes the need to consider early appropriate support interventions for not only CYP fulfilling the criteria for ADHD, but also those with subthreshold symptoms.

Subthreshold traits may also indicate a higher risk of a range of mental disorders including anxiety, depression, and suicidality. Thus a 15-year longitudinal study of subthreshold psychiatric conditions conducted with 1505 community-drawn young adults demonstrated an escalation of subthreshold NDEBID conditions to either the same (homotypic) or other (heterotypic) full-syndrome disorders of major depression, bipolar, anxiety disorders, alcohol use, substance use, conduct disorder, and ADHD [48].

Further longitudinal studies of neurodevelopmental, emotional, and behavioral symptoms from preschool to adulthood are required to help quantify the risks and cost-effectiveness of interventions for preventing dysfunctional impairments continuing in later life [49].

#### 3.5.1. Cognitive and Academic Dysfunction

Executive function impairment is a common occurrence in CYP with both a full NDD diagnosis and subthreshold traits [50,51]. This impairment may be an important mediator of the academic effects of the conditions. There is particular concern about the academic impact of subthreshold ADHD. Children with subthreshold ADHD have been found to experience significant symptoms and functional impairments [40]. Similar degrees of poor academic (reading and numeracy) and non-academic (school engagement, attendance, peer victimization, and parental expectations) outcomes were found among a group of 356 Australian school age children with subthreshold or full ADHD [52]. Another community population of children with subthreshold ADHD experienced increased risk of grade retention and graduation failure when assessed in adolescence [53]. However, in the prospective Avon Longitudinal Study over 13 years, of 11,640 children, preschool age hyperactivity/inattention and conduct problems showed significant negative effects on academic outcomes in terms of national examination results only in those where findings were suggestive of both ADHD and conduct disorder [54]. In this study, the effects of findings compatible with “borderline” ADHD did not reach significance, although “borderline” scores suggestive of conduct disorder did predict a poor outcome. Similarly, a Finnish birth cohort study of children with perinatal risk factors (e.g., low birthweight) has shown that adults who had subthreshold ADHD with attentional problems as children did not experience negative outcomes compared to those with no attentional symptoms [55].

#### 3.5.2. Psychosocial Difficulties

Studies among preschool children showed that mild-to-moderate autistic traits, described as corresponding to subthreshold autism, are associated with a significantly higher risk of emotional/behavioral problems and lower intelligence quotient compared to other children [56]. Subthreshold autism among preschool children is also associated with emotional dysregulation (e.g., aggression, anxiety/depression, sleep problems) and developmental disabilities, especially expressive and receptive language skills delay [57]. Students with autistic traits experienced different temporal sensitivity from controls, with a tendency to judge short durations as longer [58]. Autistic traits have also been linked to lower personal accomplishment, burnout, and depression among a group of Japanese medical students [59]. There is increasing interest in the use of camouflaging strategies by young people with ASD traits in coping with their social environments [23].

Balázs & Keresztény completed a systematic review of CYP with subthreshold ADHD, in 2014, which found a meaningful impact on functioning and concluded that focusing on subthreshold ADHD can be important in preventative interventions [17]. A specific mechanism that might be open to intervention emerges from a study suggesting that children with subthreshold ADHD experience impairments in facial emotion recognition, which is predictive for social and emotional problems [60].

#### 3.5.3. Crossing the Diagnostic Threshold

There is an identified risk of homotypic progression from subthreshold to full-syndrome conditions, especially in relation to ADHD. Up to 75% of adult-onset ADHD diagnoses were found to have had subthreshold ADHD scores at least at one point in childhood, suggesting progression of functional impairment with age [61]. The WHO Adult ADHD survey among 20 representative countries identified that the conditional prevalence of current ADHD averaged at 57.0% among adults with a history consistent with childhood ADHD and at 41.1% among those consistent with childhood subthreshold ADHD [38]. In their twin study of the antecedents of late-onset ADHD, Liu et al. [42] found that while low socioeconomic status predicted de novo late-onset ADHD, progression from a subthreshold ADHD status in childhood to meeting criteria for late-onset ADHD was predicted by indicators of harsh parenting and higher maternal depression.

On the other hand, we found no evidence of homotypic escalation of autistic traits. A longitudinal study of 6439 children between the ages of 7 and 13 years in the UK, showed that autistic traits are more stable over the childhood lifespan [62]. Similar findings have been reported for the persistence of autism diagnosis from childhood to adolescence, despite recorded improvements in in social interactions, repetitive/stereotyped and adaptive behaviors, and emotional responsiveness [63].

#### 3.5.4. Co-Occurring Psychiatric Disorders

Heterotypic escalation into other psychiatric disorders is well recognized in subthreshold ASD and ADHD. Normal childhood population studies have shown that subthreshold autism traits are associated with high risks for various mental health comorbidities [27]. Similar findings have been reported among preschool children with subclinical autistic traits presenting higher risks for later childhood emotional and behavioral problems [64]. Adult studies also confirm higher risks of mood symptoms and suicidal ideation both with subthreshold forms (autistic traits) (AT), and those with full-blown autism spectrum disorder (ASD) [65].

Some studies about subthreshold ASD have been conducted among adult patient populations. Matsuo et al. [66] claimed that their study of Japanese psychiatric patients compared with healthy controls was the first to show a higher degree of “autistic like traits” amongst adults with mood disorders and schizophrenia (both for remitted and unremitted conditions). They hypothesized that this represented an underlying “shared pathophysiology”. An Italian study of 138 participants meeting DSM-5 criteria for eating disorders and 160 healthy control participants (HCs), showed significantly higher prevalence of “Adult Subthreshold Autism Spectrum” among the patients compared to the controls [10]. The same research team also reported significant levels of autistic traits among 43% of subjects with bipolar disorder, with a higher risk of early onset, longer hospital stays, higher comorbid rates of anxiety, and depression and suicidality across the lifetime [29]. In a clinic sample, Memis et al. found higher scores for autistic traits among individuals with adult-onset OCD, compared to adolescents with OCD [67]. Another study of adults with fibromyalgia found that post-traumatic stress disorder (PTSD) was significantly associated with higher levels of autistic traits starting in early childhood [20]. Autistic traits have also been reported to increase the risk of depression and potential vulnerability to burnout among medical students [59]. These findings suggest a possible indication for closer monitoring of CYP with subthreshold symptoms in some settings because of an increased risk of adult-onset disorders.

A population-based twin study of 312 children showed that subthreshold diagnoses of ADHD and disruptive behaviour disorders co-existed with high scores on screening for psychiatric diagnoses including depression, mania, panic attack, phobias, anorexia nervosa, motor tics, and PTSD in girls, and with depression and PTSD in boys, as well as with smoking and high alcohol consumption in both sexes [26]. Among a group of 2493 South Korean school children, subthreshold ADHD was associated with an increased risk of externalizing disorders as well as somatic complaints, anxious/depressed, social problems, attention problems, delinquent behaviors, aggressive behaviors, externalizing problems, and total behavioral problems compared to the controls [37]. In a cohort of 4635 Swedish twins screened for ADHD symptoms at age 9 or 12, subthreshold ADHD was associated with a higher risk of negative psychosocial outcomes in adolescence, including internalizing problems, antisocial behaviour, and substance misuse [68].

Biederman et al. [18] reported on a clinic sample with 7% of children referred to a large academic center diagnosed with subthreshold ADHD after failing to meet full-threshold diagnosis for ADHD due to either insufficient symptoms or later age at onset. These children were found to have higher risks of morbidity and disability compared to healthy controls and similar risks compared to those with the full syndrome. These included the mean number of comorbid disorders, rates of mood, anxiety, and elimination disorders, substance use disorders, rates of requiring extra help in school and being placed in a special class, and elevated Global Assessment of Functioning scores. However, children with subthreshold ADHD had fewer perinatal complications, better family functioning scores, and were more likely to be female, older and to come from families of higher socioeconomic status than subjects with full ADHD [18].

### 3.6. Review of NDD Classification Conceptual Models

#### 3.6.1. Spectrum vs. Category

The classification of NDD disorders in CYP and later in adulthood has undergone considerable change in the past few decades. Of particular significance is the transition from the diagnosis of Asperger’s syndrome in the DSM-IV to its abolition in the subsequent DSM-5 [7]. A similar transition is the abolition of ADD as a specific spectrum of ADHD.

There is ongoing controversy about substituting the current model of categorical classification with a more continuous or dimensional approach. Many authors have argued that most NDEBIDs are heterogeneous and should be conceptualized as a dimensional spectrum of functional impairments, rather than unitary categories that are exclusively separate one from another [69]. A similar concept has been proposed by Bishop and Rutter [70]. They have suggested using the term “Neurodevelopmental Disability” as an alternative generic term instead of separate diagnoses, with the specific type of difficulties listed and described in more detail. With emerging evidence of a genetic overlap between several clinical NDDs, including ASD and ADHD, based on several family, twin, and molecular genetic studies, the existence of common causal pathways and possibly of a single neuropathologic entity has been suggested [71]. The hypothesis of developmental neurobiology, called neuroconstructivism, views the brain as an interacting system where disturbance in one local area in the early stages of development can have a cascading effect on a range of other cognitive domains. This encourages a more generic approach to the description and identification of NDEBIDs [72].

Some authors have highlighted the potential risks associated with expanding the scope of clinical diagnosis of NDD conditions to include subthreshold diagnoses [3]. Expanding diagnoses might be unhelpful due to potential social, psychological, and health risks, such as over-stretched demand for limited healthcare resources and elongated waiting periods for individuals with more severe symptoms gaining access to treatment [73].

An alternative model of dimensional classification of NDEBIDs and related mental health conditions being considered in the context of research is the Research Domain Criteria project proposed by the NIMH [1]. This project provides another framework of classifying mental disorders as dimensional constructs which integrate information across multiple measurement levels (e.g., genes, molecules, cells, circuits, and self-reports) [74].

The DSM-5 appears to acknowledge the value of recognizing subthreshold ADHD as it has accepted that adults and adolescents may be given a full diagnosis of ADHD even if they do not meet the full criteria for childhood symptoms. This only applies if conditions of perseverance and functional impairment in multiple settings are met. This is a desirable development from clinical assessment based on the categorical design of standard diagnostic manuals (ICD-11 and earlier versions of the DSM), which is to either give a diagnosis where full criteria are met or “no diagnosis”, even when significant traits which approach but do not cross the diagnostic threshold are present [75].

#### 3.6.2. The Concept of Neurodiversity Rather than Disorder

Despite increasing prevalence and recognition of neurodevelopmental disorders or their subthreshold traits, affected individuals are still often exposed to social stigma and pressure to conform to “normal behaviors”, with consequent risks of emotional stress and burnout [76]. The involvement of groups representing individuals with NDDs in the discourse has supported an alternative model of neurodiversity, where diagnostic categories are reframed as infinite variations within the general population that are indicators of difference rather than disorder. This way of thinking has many advantages in enabling individuals to find their place in society [77] but raises many challenges [78]. Relying on lived experience of difference, the neurodiversity approach does not follow the diagnostic boundaries of medical diagnoses and may make the concepts of threshold and subthreshold disorder redundant. The future development of classification systems will take account of the views of neurodiversity advocates and clinicians and may require further reflection on the value of diagnostic thresholds and the validity of subthreshold conditions.

### 3.7. Management

There are very few studies and only limited evidence that a person who does not meet the full criteria for a specific neurodevelopmental disorder may benefit from approaches addressing subthreshold symptoms through interventions, therapy, and support services. A Dutch prospective, multicenter study of a clinic-based sample described family functioning (cohesiveness, interaction, effective communication, decision-making and problem-solving, and getting along with each other) among 2–10-year-old children with either autism or autistic traits. The study identified over 1 year that, for the children with subthreshold autistic traits, there was a significant, unidirectional association between having fewer autism traits and better family functioning. The authors suggested that education and support need to be more integrated and personalized, focusing on the individual and family needs of non-autistic children with high subclinical autism trait levels instead of only providing this to families of autistic children [79]. A double-blind, randomized control trial of omega-3 and Korean red ginseng in children with subthreshold ADHD showed small but significant improvements in parent ratings of attentional symptoms [80]. This led the authors to suggest that small improvements may have the effect of preventing progression from a subthreshold condition to a diagnosable disorder, but without longer follow up it is not possible to comment on the impact of dietary supplements.

The biopsychosocial model implies that the significance of milder symptoms and subthreshold NDD categories requires special attention from all neurodevelopmental clinicians, as well as education and social practitioners, to ensure that all the vulnerable CYP coming under this category have their additional mental and psychosocial needs identified and appropriately supported [77]. However, the identification of unrecognized conditions should be driven by a sound understanding of the advantages and potential disadvantages of this approach and there is still limited evidence in either direction.

If there is a possibility that the presence of an unmitigated subthreshold condition is a stressor that contributes to the etiology of another psychiatric disorder, rather than simply a sign of a shared pathophysiology, the need to remediate the subthreshold condition may become part of the treatment of the associated disorder. It is also relevant to consider the possibility that a subthreshold condition may have an impact on the efficacy of the treatment process. Thapar et al. [3] suggest a “conceptual approach” in clinical situations, where a biopsychosocial formulation identifies subthreshold NDD as an etiological factor in a child’s condition. There may be a strong case for targeting an intervention towards this component of presentation as part of a treatment package. For example, in children, this may imply a school-based approach to a subthreshold disorder affecting daily functioning even though there is a different formal diagnosis based upon impairment that arises in other settings. Thapar et al. [3] also consider the clinical implications of a developmental overview, taking account of subthreshold conditions alongside formal diagnoses in planning clinical interventions over time.

It might be suggested that most individuals with subthreshold symptoms would benefit from either self-referral or professional support with non-pharmacological behavioral and psychosocial interventions [81], but there are reasons for caution, not least considering the large numbers involved and their potential impact on services for the more severely impaired [78]. When the symptoms escalate or there is deterioration of functional impairment, pharmacological interventions may be added to the support being offered. Thapar et al. [3] argue that individuals with subthreshold symptoms but substantial impairment should not be allowed to miss out on appropriate service provision. They suggest an urgent need for the development of systematic and validated methods for assessing the needs and degree of multi-domain functional impairment of CYP with neurodevelopmental disorders beyond any diagnostic criteria they might fulfil. Garralda [82] has proposed a future development of expanded early neurodiversity diagnostic/subsyndromal support services, run by child and adolescent mental health and pediatric services collaboratively, that might meet some needs.

The neurodiversity movement advocates for an alternative approach, suggesting reasonable adjustments in societal environment structures that allow people with distinct neurodevelopmental conditions or traits to thrive in their own way, rather than managing symptoms or trying to make them “behave normally” [76]. A pilot study using key concepts from the neurodiversity approach (encouraging better understanding, early support, and environmental changes) for neurodiverse children showed significant improvements in the general wellbeing of the affected individuals [83].

### 3.8. Raising Public Awareness

The general public need better information and education about the wider prevalence and meaning of NDD disorders (including ADHD and autism) even at a subthreshold level. There are good reasons for concern about medicalization and in the absence of evidence to support interventions there may be more value in inclusive approaches. Although there is controversy around concepts of neurodiversity [77], this model may offer a better understanding of the complexity of individual differences at a subthreshold level. The inclusion agenda is becoming more visible within the workplace, which offers the opportunity for education of working-age adults. Campaigners might argue that the best way to reduce the impact of subthreshold conditions is to change the social environment, and the workplace is an obvious setting to effect such change. For example, a study of medical students might be used to argue for changes in medical training programs to address features that may challenge a neurodiverse individual and, thus, improve outcomes for those with subthreshold ASD rather than focusing on an individual student’s vulnerabilities [59].

It is important to consider cultural and contextual factors when assessing and diagnosing neurodevelopmental disorders. Brosco & Bona [84] have shown that rapidly increasing rates of ADHD diagnosis in children and adolescents in the United States coincide with increasing academic demands in young children. Banaschewski et al. [85] have highlighted the finding that, although measures of attention and overactivity in populations suggest that the prevalence of ADHD is similar in countries across the globe, the rates of diagnosis vary considerably from country to country and appear to be heavily influenced by cultural factors. Every society provides the context for either supporting or disabling the individuals with diversified personality or behavioral traits, and the impairment criterion required for a diagnosis will vary with the context. In addition, a range of social factors may influence the perception of personality traits as either normal or abnormal. Banaschewski et al. [85] have argued that the recent “push for performance” in the post-industrial era in the Western world might explain and predict the increasing prevalence of ADHD and a similar process might apply in relation to other neurodiversity diagnoses. Other possible social factors influencing rates of diagnosis include changes in schooling practices and media coverage, increased public and clinician awareness and acceptance, and reduced stigmatization. It has been argued that the concept of NDD could in itself contribute to difficulties faced by mildly affected individuals, and so public education should focus on difference not on disorder [86]. One might expect the drive for medicalization to reduce if society were more tolerant and supportive of people with neuro-differences [85]. It is worth considering the potential impact of such factors in relation to the effect of subthreshold conditions.

### 3.9. Future Research Directions

The concepts of subthreshold disorders, conditions, and traits are widely recognized but not well delineated. For future research to be widely applicable there needs to be agreement on the precise meaning of these terms. This way forward for research might be developed most effectively with the involvement of individuals and families for whom the recognition of subthreshold conditions has personal meaning, within a cultural context. In view of the uncertainty that remains about the association of subthreshold conditions with negative outcomes, there is a great need to explore the possibility that features of different cultural and educational settings may affect their salience. This would enable evidence-based investigation of the proposal that the primary therapeutic target according to the neurodiversity concept should be to change the context. Such a change should be designed in a way that neurodiverse people can have positive and affirmative experiences and draw on their resources and specific characteristics [85].

Specific NDD traits questionnaires for CYP need to be developed and evaluated for validity and reliability, rather than using the modified adult versions. Further studies of the neurobiology of CYP with subthreshold clinical traits of NDEBIDs compared to those with full-clinical diagnosis and controls would greatly help us to improve our knowledge for identification and management of those affected [44].

Screening for subthreshold symptoms should be evaluated among a cohort of individuals with a high risk of developing of one or more NDEBIDs. For example, up to one-third of 22q11.2 deletion syndrome individuals are known to have a risk of developing psychosis, usually preceded by subthreshold negative symptoms of psychosis, and such individual traits in social functioning may be identified early through close monitoring [87,88]. Probands and siblings of CYP with a full diagnosis of NDD are known to be at higher risk of subthreshold symptoms [41]. This might be a strong indication for wider screening of family members among CYP with NDD, as well as prevention and intervention efforts that aim to alleviate the negative downstream consequences associated with disorders of neurodevelopment.

Some neurodevelopmental disorders may not manifest fully until later in development, while others may improve with appropriate supportive interventions in multiple settings. Longitudinal assessment, i.e., tracking symptoms over time, may be necessary to identify and understand the progression of these conditions.

More research is urgently required to identify the effectiveness of multi-domain supportive strategies for CYP with subthreshold NDD conditions, which would help to address the public health challenges they may present. As research is likely to lead to the identification of greater numbers of individuals with subthreshold conditions, there is an urgent need to explore the cost and value of various forms of interventions in this population, considering both the alleviation of current impairment and the reduction in lifetime risks associated with these conditions. The suggestion by Liu et al. [42] that parenting behaviour and parental mental health problems may promote the development of late-onset ADHD in children with the subthreshold condition implies the potential value of research focusing on intervention with families. A biopsychosocial model of support has been suggested as invaluable for all individuals with any neurodevelopmental traits, even if they do not have a specific diagnosis [77], but there is a need for further consideration of the implications of this approach, including its impact on resource allocation. Alternative approaches based upon an inclusion model in schools, workplaces, or more generally in society, require rigorous assessment, and their benefits need to be weighed against potential unforeseen consequences and costs.

## 4. Discussion

This review sets out to identify practical implications from the literature of subthreshold ADHD and ASD. There are many potential benefits of greater emphasis and understanding of subthreshold NDD symptoms, with implications for clinicians dedicated to carrying out comprehensive emotional and developmental assessment of all CYP presenting with neurobehavioral difficulties, but there are also grounds for caution, particularly as there is no shared definition of the subthreshold concept. Some studies failed to identify the significant effects on outcome of subthreshold ADHD but, in general, there is further evidence to support the view (particularly in relation to ASD) that the identification of subthreshold conditions could avoid delayed or missed opportunities for diagnosis and treatment, which could otherwise be accompanied by significant negative psychopathological consequences. This concept of a wider comprehensive assessment would also accommodate early identification of multiple comorbidities as well as subthreshold presentations [69]. The categorical approach of restricting diagnosis, with thresholds always having to be met, seems deficient as it does not recognize the significant impact of neurodivergent traits on the daily functioning of the individuals and underestimates the social, educational, emotional, and occupational distress that they may experience if subthreshold conditions are not recognized and appropriately supported. Therefore, this paper argues for further review of the current diagnostic paradigm and its application within a societal context.

In view of significant functional impairments in children and adults with subthreshold symptoms of various NDEBIDs, many researchers have advocated for a dimensional subthreshold diagnostic category for each condition [43]. Several studies have concluded that a broader clinical assessment is needed for CYP with any NDEBID-related symptoms [26]. The potential associations of subthreshold ASD or ADHD with other psychiatric disorders may be a consequence of shared pathophysiology and/or of the impact of adverse experience upon a developmentally vulnerable individual but, in any case, the recognition of this association gives weight to the argument for further assessment when a subthreshold condition is identified.

Despite the limitations of research and lack of consensus on definitions, there are practical implications of this review in health and education. We need a new approach to how CYP with NDD difficulties are supported. For example, subthreshold levels of ADHD symptoms typically do not qualify affected students for special educational interventions and yet increase the risk of adverse educational outcomes [53]. A child whose symptoms do not meet all the diagnostic criteria may require additional psychosocial and academic support in several settings. There should be a way of allocating resources to take account of the needs of children with all levels of difficulty. There is a case for an approach based upon principles of neurodiversity, especially in relation to subthreshold conditions. The input of affected individuals should be a core commitment in research and policy making.

For the clinician, the traditional approach of “all or none” in terms of diagnosing NDEBIDs needs to be reviewed and possibly abandoned. The identification of any neurodevelopmental impairment should be seen as a starting point for the further assessment of other domains rather than being an endpoint of a unitary diagnosis that ignores many other significant difficulties [89]. The case for such a comprehensive assessment is particularly strong where individuals present clinically with significant impairments and complex neurodevelopmental profiles. Similarly, mental health assessment should take account of the implications of co-occurring subthreshold conditions. Recognition of such components of a clinical case formulation may contribute to a better understanding of the way forward, but where there is no evidence to support intervention there needs to be caution that the identification of a subthreshold condition does not imply that therapeutic or remedial action will necessarily follow in clinical settings.

## 5. Conclusions

Subthreshold traits (insufficient symptoms to make a diagnosis but some evidence of impairment) are common in the general population and are strong predictors of poorer adult mental health and functional outcomes [45]. While subthreshold symptoms may not meet the full criteria for a specific neurodevelopmental disorder, they can still have significant implications for an individual’s wellbeing and functioning.

Greater clarity regarding the status of subthreshold conditions is required to support a research agenda that should be developed with public involvement and in dialogue with proponents of neurodiversity models. Further longitudinal studies are needed to identify the cost-effectiveness of early identification and interventions for CYP with subthreshold neurodevelopmental conditions in different settings over a lifetime.

We argue that a “subthreshold condition” should be recorded for any CYP with features of NDDs that do not meet the categorical DSM-5 or ICD-11 criteria, if there is evidence of significant impairment persisting in at least one setting. This would be particularly relevant where there is a significant psychiatric disorder that may be ameliorated by attention to the vulnerabilities identified in association with subthreshold NDDs. A thorough and nuanced assessment, considering the spectrum nature of these disorders, is desirable to guide appropriate interventions.

## Figures and Tables

**Figure 1 pediatrrep-17-00042-f001:**
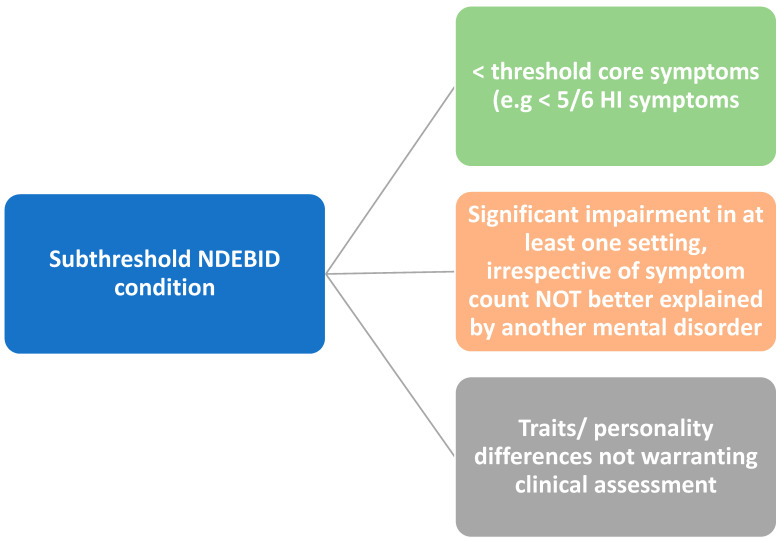
Some conceptualizations of subthreshold NDEBID conditions.

**Table 1 pediatrrep-17-00042-t001:** Summary of prevalence studies for various subthreshold conditions.

Authors	Population	ADHD Traits	ASD Traits
[35] Ek et al., 2007	General population	1.6%	
[37] Cho et al., 2009	Korean school children	9%	
[36] Larsson et al., 2012	9–12 yr Swedish twins	9.75%	
[17] Balázs & Keresztény	LR	0.8–23.1%	
[31] Arildskov et al., 2016	OCD		10–17%
[38] Fayyad et al., 2016	Adults	3.7%	
[34] Kirova et al., 2019	LR/Meta-analysis	17.7%	

Legend: LR—literature review; OCD—obsessional compulsive disorder.

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
