# Peer review of "Subthreshold Autism and ADHD: A Brief Narrative Review for Frontline Clinicians"

_pediatrrep, 2025, doi:10.3390/pediatric17020042_

Round 1
Reviewer 1 Report
Comments and Suggestions for Authors
The authors of this review are essentially advocating for a new approach to diagnosing neurodevelopmental disabilities, advocating for the development of a dimensional approach rather than a categorical approach. They correctly point out the risk of "medicalizing" behavioral traits that may merely be deviations from typical that are not disruptive to function and the question of access to services if the definition of these conditions is broadened.
My major concern is that the authors don't distinguish between subthreshold ADHD based on number of symptoms vs. the presence in 2 or more settings. The requirement for 2 or more settings is valuable in distinguishing ADHD from a learning disability (hence only present in school) or a difficulty home situation (hence only present at home; I am reminded of a child I saw doing fine in school but brought in for hyperactivity and ? of ADHD and the history revealed that the 4 family members were living in a studio apartment. My diagnosis was cabin fever rather than ADHD). In the discussion of co-morbidity the authors don't seem to take into account the last part of the diagnostic criteria for ADHD:
- The symptoms do not happen only during the course of schizophrenia or another psychotic disorder. The symptoms are not better explained by another mental disorder (e.g. Mood Disorder, Anxiety Disorder, Dissociative Disorder, or a Personality Disorder. If there are co-morbid mental disorders then it may well NOT be ADHD and treatment should be directed at the mental disorder.
Similarly in the case Autism. If not meeting all the criteria is it "subthreshold autism" or is it OCD (excessive rigidity without the social imparment), or pathological shyness/anxiety causing withdrawal without the rigidity). There is a reason that the presence of Persistent deficits in social communication and social interaction across multiple contexts AND Restricted, repetitive patterns of behavior, interests, or activities is required. As in the case of ADHD discussed above, if both are present but not quite at threshold that is distinct from one category being present and the other not.
Finally, the. authors don't sufficiently emphasize another criterion for both conditions:
There is clear evidence that the symptoms interfere with, or reduce the quality of, social, school, or work functioning for ADHD
Symptoms cause clinically significant impairment in social, occupational, or other important areas of current functioning for Autism
If some criteria are met but not disrupting function, that's not a subthreshold condition, that's biological variabiiltiy.
Finally, I'm not sure who the target audience is, but the paper is rather long and would likely find a broader readership if shortened.
Reviewer 2 Report
Comments and Suggestions for Authors
The topic of this manuscript is very interesting. The authors provide a narrative review about subthreshold NDD, focusing mainly on ADHD and ASD. The aim is to identify any practical lessons that may be applicable to frontline neurodevelopmental clinicians.
The introduction provides a good overview of the topic. Also, the results section is well written and discussed. The authors, through the review of the current literature, try to define the state of the art of the definition and classification of Subthreshold ASD and ADHD, and their social and clinical implications. However, further studies and a general consensus about these conditions are needed.
Minor:
- The paragraph "3.10. Figures, Tables and Schemes" is unnecessary. The figures and tables are mentioned in the main text.
-In the abstract and the main text, there are some typos. Please, correct them.
Reviewer 3 Report
Comments and Suggestions for Authors
Dear Authors,
Thank you very much for allowing me to review your article. I have read it carefully and it is, in general terms, very well written. However, I propose some changes before being considered for publication.
Abstract
The abstract seems to me to reflect well the content of the article. I have no suggestions.
Introduction
The introduction clearly explains the problem to be addressed and the existing knowledge gap in the current literature. The aim of the research is significant. Indeed, a narrative review to evaluate publications related to neurodevelopmental disorders, focusing mainly on attention deficit hyperactivity disorder and autism spectrum disorder is relevant and necessary. In addition, the article has a clear practical orientation which is unusual. In fact, the practical implications that can be applied as effective public health measures are especially important. Overall, very good work.
Suggestions: If the abbreviation DSM V refers to the Diagnostic Manual of Mental Disorders it is important to specify it.
Materials and Methods
Suggestions: It is important that the authors explain the differences and also the advantages that a narrative review may have over a systematic review. Differentiating between the two types of reviews will enrich this section much more.
Results
Definitions
The definitions are detailed and comprehensive. The authors make a great effort to ensure that the reader fully understands the terms used.
Suggestions: It is strange that the authors cite Figure 1 and it does not appear immediately following the citation. This is an issue for the authors to discuss and think about.
Assessment Tools
The evaluation tools are adequately described. I have no suggestions.
Prevalence
The section is very well written and clear.
Suggestions: Exactly the same as with figure 1.
Risk Factors and Neurobiology
The section is clear and contains sufficient information. I have no suggestions.
Lifetime Functional Impairment
Cognitive and Academic Dysfunction
Psychosocial Difficulties
Crossing the Diagnostic Threshold
Co-occurring Psychiatric Disorders
The different sections and subsections are developed in great detail. I have no suggestions.
Review of NDD Classification Conceptual Models
Spectrum vs Category
The Concept of Neurodiversity rather than Disorder
The different sections and subsections are developed in great detail. I have no suggestions.
Management
Raising Public Awareness
These two sections are very well developed. I have no suggestions.
Future Research Directions
The authors propose very interesting future lines of research.
Suggestions: This section together with the limitations of the study is usually placed after the discussion of results and practical implications, just before the conclusions.
Figures, Tables and Schemes
Suggestions: As I said before, figures and tables should be included just after they are mentioned. In addition, the figure and table should improve their presentation.
Discussion
Suggestions: Usually, the discussion begins with the objective of the research and from this objective all the results obtained are discussed.
I would like the authors to include a subsection with relevant practical implications and indications of how to carry them out.
Conclusions
Overall, these are good conclusions. I have no suggestions.
Comments on the Quality of English Language
I have no comments regarding the quality of the English language.
Round 2
Reviewer 1 Report
Comments and Suggestions for Authors
This is an improvement but the issues raised previously remain. I feel rather strongly that "The presence of symptoms/impairment in one but not in other settings" is much more likely to be a problem in the setting (learning disability if at school; parenting or other domestic issue if at home) than a subthreshold condition and I believe that should be removed as a basis for calling something "subthreshold.
"uncontrolled pilot study involving early identification of subthreshold symptoms of au- 463 tism in twelve children with no ASD diagnosis with appropriately targeted interventions 464 reported improvement in specific outcomes such as reduced children's screen-time, repet- 465 itive behaviours decreased and EEG ratio power in some channels [79]"
An uncontrolled study involving 12 chldren. This is worthless and should be deleted.
The United Nations Convention on the Rights of the Child defines child as, "A human being below the age of 18 years unless under the law applicable to the child, majority is attained earlier."
Can't we replace CYP with child/children?
Demontis, D., Walters, G.B., Athanasiadis, G. et al. Genome-wide analyses of ADHD identify 27 risk loci, refine the genetic architecture and implicate several cognitive domains. Nat Genet 2023; 55: 198–208. https://doi.org/10.1038/s41588-022-01285-8
Thapar A. Discoveries on the genetics of ADHD in the 21st century: new findings and their implications. Am J Psychiatry 2018;175(10):943-50. PMID 30111187
Powell V, Martin J, Thapar A, Rice F, Anney RJL. Investigating regions of shared genetic variation in attention deficit/hyperactivity disorder and major depressive disorder: a GWAS meta‑analysis. Sci Rep 2021;11(1):7353. PMID 33795730
These are 3 studies showing overlap of ADHD candidate genes with other NDD, psychiatric and cogntive candidate genes and might be supportive of the suthors' thesis.
Author Response
Please the attachment.

Reviewer 3 Report
Comments and Suggestions for Authors
Dear Authors,
Thank you very much for allowing me to review this second version of your article. I have carefully read all the changes and the article, in general, has improved. The authors have eliminated certain abbreviations which improves the overall readability. The authors include the advantages of a narrative review over a systematic review which adds value to the article. The authors provide a convincing answer as to where they place future lines of research. The authors reorganize the discussion of results, and the practical implications are clearer. The article is ready for publication.
Author Response
Comment 1:
Thank you very much for allowing me to review this second version of your article. I have carefully read all the changes and the article, in general, has improved. The authors have eliminated certain abbreviations which improves the overall readability. The authors include the advantages of a narrative review over a systematic review which adds value to the article. The authors provide a convincing answer as to where they place future lines of research. The authors reorganize the discussion.
Response 1
We appreciate the positive comments of Reviewer 3.